# A Review on Carbon Quantum Dots Modified g-C₃N₄-Based Photocatalysts and Potential Application in Wastewater Treatment

**Shilpa Patial** [1]**, Sonu** [1]ORCID**, Anita Sudhaik** [1]**, Naresh Chandel** [2]**, Tansir Ahamad** [3]**, Pankaj Raizada** [1,*]**, Pardeep Singh** [1]**, Nhamo Chaukura** [4]ORCID **and Rangabhashiyam Selvasembian** [5,*]ORCID

1. School of Advanced Chemical Sciences, Shoolini University, Solan 173212, Himachal Pradesh, India
2. Department of Chemistry, RNT Government College, Sarkaghat, Mandi 175024, Himachal Pradesh, India
3. Department of Chemistry, College of Science, King Saud University, Riyadh 11362, Saudi Arabia
4. Department of Physical and Earth Sciences, Sol Plaatje University, Kimberley 8301, South Africa
5. School of Chemical and Biotechnology, SASTRA Deemed University, Thanjavur 613401, Tamil Nadu, India

\* Correspondence: pankajraizada@shooliniuniversity.com (P.R.); rambhashiyam@gmail.com (R.S.)

**Abstract:** Carbon quantum dots (CDs) are a fascinating class of carbon nanomaterials (less than 10 nm in size) with unique optical, electrical, and physicochemical properties. In addition to these properties, CQDs exhibit the desired advantages of aqueous stability, low toxicity, high surface area, economic feasibility, chemical inertness, and highly tunable photoluminescence behaviour. Recently, graphitic carbon nitride (g-C₃N₄) has appeared as one of the required stable carbon-based polymers due to its varied applications in several fields. In this regard, modification strategies have been made in the g-C₃N₄ semiconductor using CQDs to enhance the adsorptive and photocatalytic activity. In comparison to other semiconductor quantum dots, g-C₃N₄ shows strong fluorescent properties, such as wide excitation spectra, photostability, and tunable photo-luminescent emission spectra. The interaction inside this multicomponent photocatalyst further promotes the photocatalytic activity by improving charge transference, which plays a vital role in electrochemistry. Therefore, CQDs are auspicious nanomaterials in the field of photocatalysis, wastewater treatment and water adsorption treatment. This particular article featured the recent progression in the field of CDs/g-C₃N₄-based photocatalysts focusing on their luminescent mechanism and potential applications in wastewater treatment.

**Keywords:** g-C₃N₄; carbon quantum dots; nanocomposite; pollutant degradation; CO₂ reduction



## 1. Introduction

With rapid urban and societal development, environmental problems, such as organic contaminants, pathogenic micro-organisms, heavy metal ion effluence and energy demands, have posed serious threats to the sustainable growth of human life [1,2]. Predominantly, wastewater is the key source of water pollution, occurring particularly from chemical industrialization, which is carcinogenic and enormously poisonous in nature due to the presence of extensive large organic complexes [3]. The conventionally utilized environmental remediation processes, such as chemical oxidation, adsorption, incineration, and biological oxidation, have been employed in the mitigation of all types of toxic and organic water contaminants [4–7]. However, the performance of these techniques is not effective, owing to an incomplete degradation of the persistent organic pollutants and noxious sludge formation [7–9]. Solar light-driven photocatalytic technologies offer a prominent strategy to resolve worldwide environmental issues [10–12]. As a novel approach to beneficiate solar energy, semiconductor-based photocatalysis provides a sustainable and environmentally friendly technology, compared to chemical and biological techniques [13]. Since the 1970s, TiO₂ has been found as an outstanding semiconducting material by Fujishima and Honda [14]. Carey and co-workers studied the photocatalytic degradation ability of the TiO₂ towards organic toxins in the aqueous solution, which increased the interest

of researchers on TiO$_2$ photocatalysis [15–17]. Thereafter, a number of semiconductors photocatalysts, involving TiO$_2$ [18], SrTiO$_3$ [19,20], black phosphorus [21], ZnO [20,22,23], and Bi$_2$MO$_6$ [24], have been fabricated and broadly studied with the application in organic pollutants photodegradation [25,26], H$_2$ evolution [27], and CO$_2$ reduction into solar fuels [28,29]. Additionally, Malato et al. summarized the research related to the solar light-assisted photodegradation of persistent water pollutants through the generation of $^\bullet$OH free radicals [30]. However, a potential semiconductor photocatalyst should have characteristic features, including suitable band-edges positions, appropriate band energy, high surface area, and more electron-hole pair separation and mobility as delineated in (Figure 1) [31]. These desirable features, such as high surface area due to the formation of nanoparticles, leading to high adsorption, efficient charge separation due to preferential migration along a certain direction, and suitable band-positioning, fulfilling the requirement of $^\bullet$OH, $^\bullet$O$_2{}^-$ free radicals [32,33].

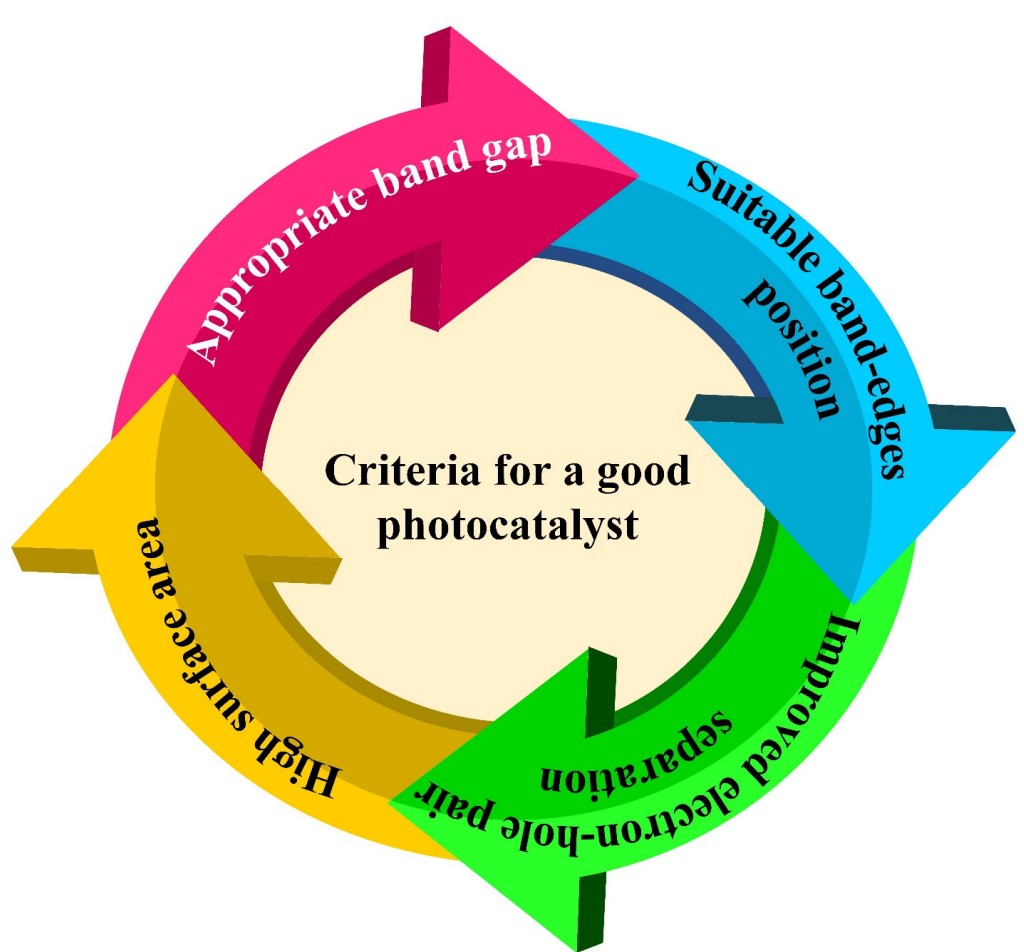

**Figure 1.** The main features of a potential photocatalyst. Adapted from Ref. [31].

In recent years, many unprecedented pieces of research were found in the field of environmental and wastewater treatments through advanced oxidation technologies (AOTs), as one category of chemical methods, which have been indicated as one of the most alluring and effectual solutions for the degradation of organic pollutants [34,35]. The merits of these processes relative to other conventional methods are due to (i) complete degradation of the persistent complexes and their intermediate products without generation of any secondary by-products, (ii) cost-effectiveness, and (iii) optimum temperature and pressure conditions being easily determined [9,36]. In particular, these processes generated highly reactive oxygen species (ROS), such as $^\bullet$OH, $^\bullet$O$_2{}^-$, $^1$O$_2$, $^\bullet$SO$_4{}^-$, and h$^+$, when reacted with H$_2$O/$^-$OH or dissolved oxygen, facilitating the degradation of nonbiodegradable organic compounds into simpler molecules (H$_2$O and CO$_2$) [37]. The strong oxidizing capability of ROS and

holes proficiently mitigate developing pollutants and boost the reaction kinetics [38]. The production of the superoxide radical ($\bullet O_2^-$) as a result of the reaction between oxygen and photoexcited electrons in the conduction band (CB) of a semiconductor and the production of hydroxyl radicals ($\bullet OH$) after a reaction with valence band (VB) holes is illustrated by the following Equations (1) and (2) [39];

$$e_{CB}^- + O_2 \rightarrow \ \bullet O_2^- \tag{1}$$

$$h_{VB}^+ + H_2O \rightarrow \bullet OH + H^+ \tag{2}$$

## 2. g-C$_3$N$_4$ as a Potential Photocatalyst

Recently, a robust and visible-light-driven semiconductor, graphitic carbon nitride (g-C$_3$N$_4$) with a characteristic two-dimensional structure has attained great attention from researchers in field of photocatalysis due to its unique properties [40–42]. These properties comprise of earth-abundance, high physicochemical stability, appealing electronic band structure, and facile synthesis methods using low-cost precursors, i.e., melamine, cyanamide, thiourea, etc., [43]. Wang et al., in 2009 reported g-C$_3$N$_4$ as a metal-free conjugated semiconductor photocatalyst when employed for photocatalytic water splitting under visible-light illumination [44]. In addition, g-C$_3$N$_4$ not only accompanied suitable band energy of 2.7 eV with 1.07 V CB position but also possesses high chemical and thermal stability because of its tri-s-triazine structure and high degree of polymerization. However, its practical applications are strictly hampered due to its inherent drawbacks, such as lower visible-light range, smaller specific surface area, lower quantum yield, and high rate of recombination due to weak van der Waals interactions between adjacent carbon nitride layers [45,46]. Therefore, researchers have focused more on altering g-C$_3$N$_4$ with fine surface and electronic characteristics via morphology control and band assembly engineering. In order to improve the photocatalytic efficiency of the pristine g-C$_3$N$_4$, various modifications have been employed, including doping, heterostructure formation, and incorporation of carbonaceous nanomaterials, such as reduced graphene oxide (rGO), carbon dots (CDs), and carbon nanospheres [47,48].

In addition, quantum dots have attained a key research focus due to their exclusive size-dependent characteristics and definite optical and electronic properties. An incipient zero-dimensional (0D) carbon quantum dots (CQDs) nanomaterial exhibits an average size of less than 10 nm. Coupling with CQDs rather than other semiconductor quantum dots is a feasible procedure to enhance the photocatalytic activity of g-C$_3$N$_4$, owing to physiochemical durability, photochemical stability, water solubility, non-toxicity, ultrasmall dimension, and photoinduced electron transport properties [49–51]. Additionally, their excellent up-conversion photoluminescence (PL) characteristic as a result of multiple-photon excitation, indicated maximum visible-light absorption [52]. It has been noticed that they possess excellent electron reservoir/transport features and distinct emission traps due to quantum confinement effect with $\pi$ conjugate structure. The usage of metal-free CQDs as remarkable co-catalyst generates a key basis for interface engineering of g-C$_3$N$_4$ photocatalyst. Therefore, the g-C$_3$N$_4$/CQDs based nanocomposites have some diverse merits over the bare g-C$_3$N$_4$, such as broader visible-light absorption, ultra-high specific surface area, and enhanced electron-hole pair migration and separation efficiency [53].

This review provides the summary of the latest research progression in the coupling of g-C$_3$N$_4$ with CQDs and other semiconductors for the preparation of new nanocomposites. The growth of number publications as a function of the year for CQDs/g-C$_3$N$_4$-based photocatalysts for wastewater treatment evaluated using the keyword 'CQDs/g-C$_3$N$_4$ + wastewater' for the last ten years, as shown in Figure 2. Firstly, we introduced the design consideration of the g-C$_3$N$_4$/CQDs-based nanocomposite. Next, we will provide a detailed discussion on the recent developments of these nanocomposites in the field of photocatalysis, particularly for the treatment of organic contaminants in water. Subsequently, these nanocomposites have been employed for the photodegradation of pollutants, including organic dyes and different antibiotic and phenolic pollutants. Finally, the ongoing

challenges and future perspectives requiring research focus along with future direction for the development of g-C$_3$N$_4$/CQDs-based nanocomposites in photocatalytic applications has been proposed.

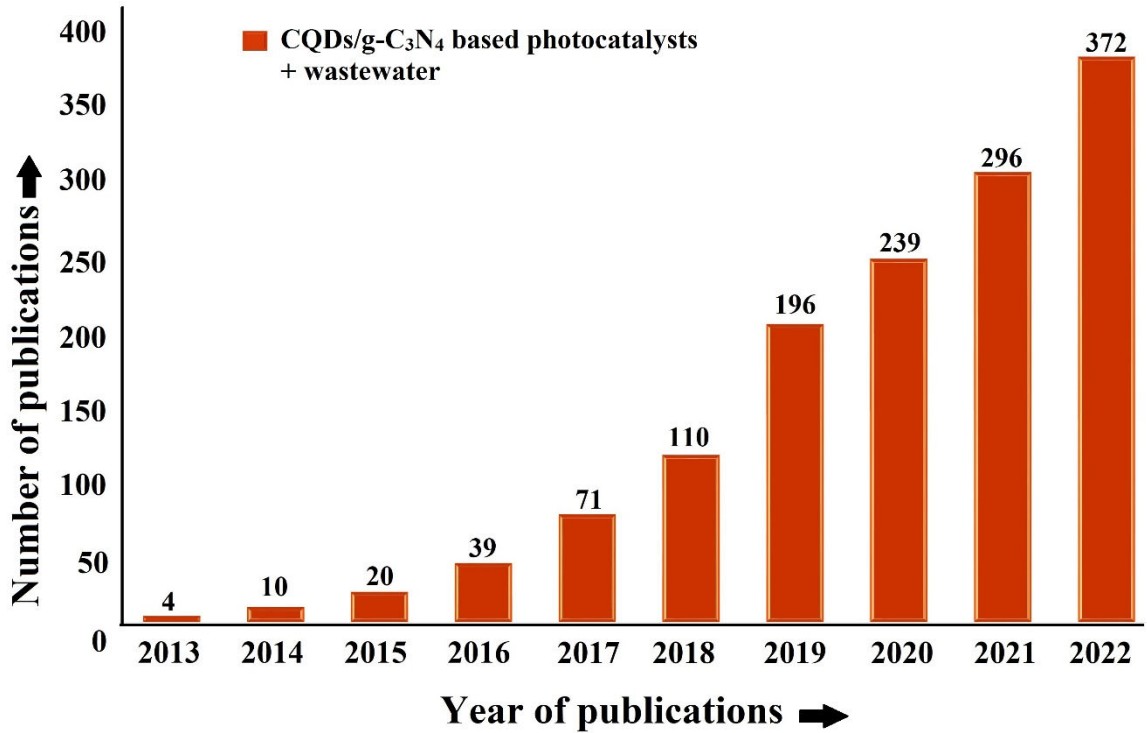

**Figure 2.** Overall publication detail on CQDs/g-C$_3$N$_4$-based photocatalysts for wastewater treatment from 2013 to 2022.

### 3. Design Consideration for g-C$_3$N$_4$/CQDs-Based Nanocomposites

To develop effectual g-C$_3$N$_4$/CQDs-based nanocomposites working for improved performance, variant crucial points must be considered. CQDs provide diverse roles when adhered onto the g-C$_3$N$_4$ surface in the field of photocatalysis application [54]. These quantum dots not only behave as photosensitizer because of photoluminescent effect and electron acceptor or mediator to direct the flow of charge-carriers but also serve as semiconducting material to generate the photogenerated electron-hole pairs, as delineated in Figure 3 [55]. Furthermore, the quick generation of photoexcited electron–hole pairs as a result of visible-light absorption range increases recombination rate of charge pairs. In the g-C$_3$N$_4$/CQDs-based heterostructure, CQDs have been employed to hamper the rate of electron–hole pair recombination, resulting in enhanced photocatalytic performance (Figure 3a). Therefore, due to their exceptional multi-photon illumination characteristic, CQDs are also functionalized as spectral convertor. The up-conversion PL properties indicated that PL emission wavelength is always inferior to the practical excitation wavelength. Additionally, lights with minimal wavelength could be achieved from higher wavelengths to obtain charge carrier excitation, increasing the photocatalytic efficiency of CQDs (Figure 3b). In addition, the CQDs also serve as electron mediators at the surface of the nanocomposite when they are either in the mode of type-II heterostructure or Z-schemes. In the charge transfer mechanism route, CQDs could behave best with an electron mediator, owing to its exceptional electron donor/acceptor characteristics, facilitating improved photocatalytic activity (Figure 3c,d) [56,57].

In consideration of multifarious mechanism of distinct g-C$_3$N$_4$/CQDs-based heterostructure can create significant advances in the photocatalytic applications. In reality,

the joined CQDs with g-C$_3$N$_4$ composites has a crucial role in the photodegradation of variant contaminants and CO$_2$ reduction, which has been elaborated in the next sections.

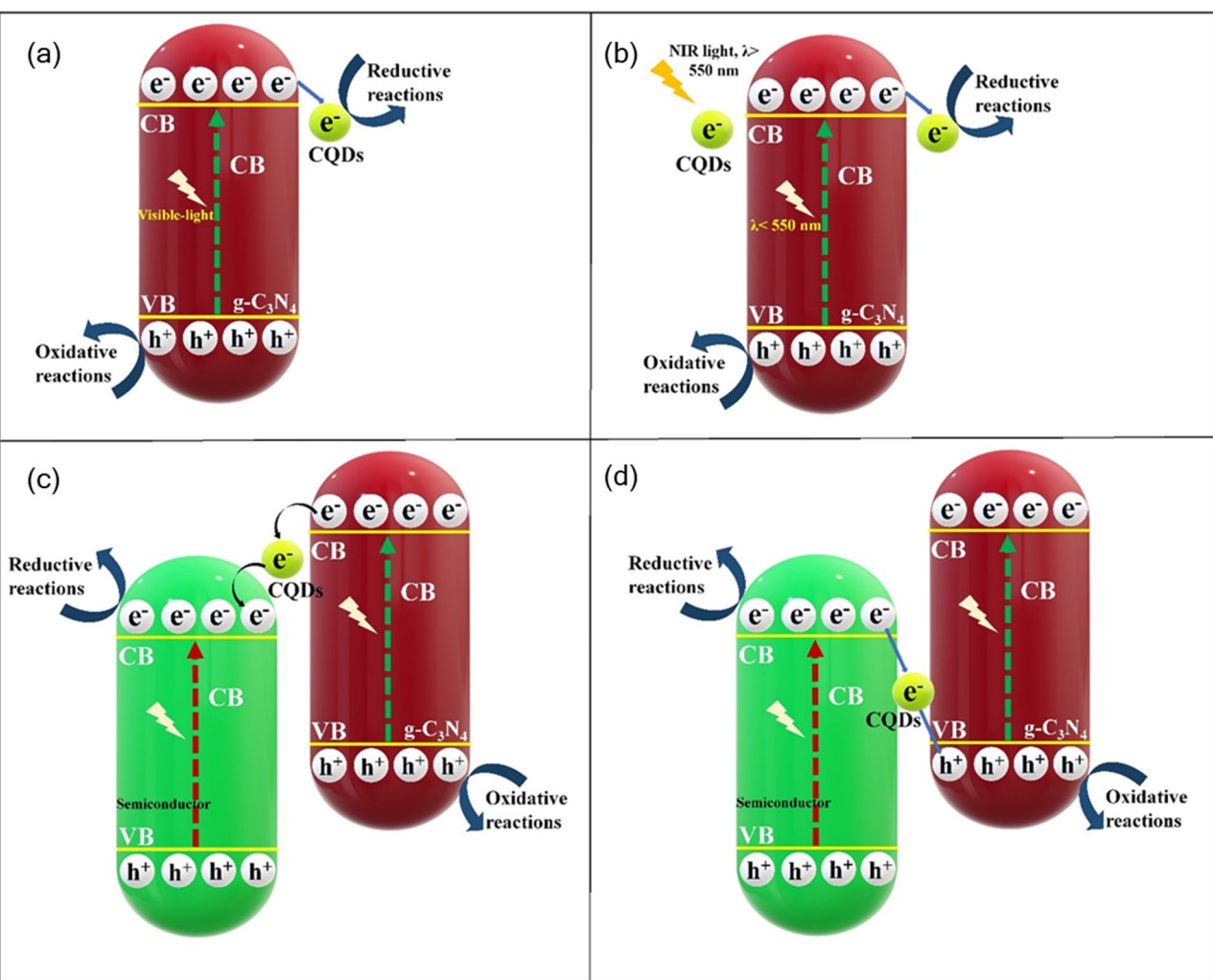

**Figure 3.** Schematic representation of the role of CQDs in the g-C$_3$N$_4$/CQDs-based nanocomposite, (**a**) CQDs hampers electron-hole pair recombination, (**b**) CQDs enables photoexcitation of electrons even with 550 nm incident light, CQDs serve as electron mediator in (**c**) type-II, and (**d**) Z-schemes. Adapted from Ref. [23].

### 3.1. g-C$_3$N$_4$/CQDs-Based Nanocomposites for Pollutant Degradation

The fast-growing urbanization and agricultural industries have led to inevitable discharge of the large quantity of hazardous, toxic, and interminably pollutants into the natural environment. Hence, with the goal of conserving the environment and apprehending the sustainable advances of human life, the photodegradation of pollutants is a research focus worldwide. In this regard, g-C$_3$N$_4$/CQDs-based nanocomposites, as novel heterogeneous photocatalysts, have recently attained great research attention due to their potential application in the eradication of a wide-range of contaminants [58,59]. For instance, Fang and co-workers studied the photocatalytic performance of CQDs-decorated g-C$_3$N$_4$, synthesized by a novel method using CQDs and dicyandiamide as chemicals, employed for rhodamine B (RhB) elimination in the presence of the UV light region [60]. This study demonstrated the photoactivity of g-C$_3$N$_4$/CQDs nanocomposite (0.013 moL$^{-1}$min$^{-1}$) to be three times greater than bare g-C$_3$N$_4$ (0.0044 moL$^{-1}$min$^{-1}$). In another piece of research, Zhang and his peer group combined g-C$_3$N$_4$ with distinct mole fractions of CQDs to evaluate the photocatalytic activity for photodegradation of phenol under visible-light irradiation [61].

The nanocomposite was prepared through feasible impregnation thermal technique and composite containing 0.5 wt% of CQDs, displaying a 3.7 times higher rate constant with best photodegradation ability than bare g-$C_3N_4$ (Figure 4a). The proposed mechanism of the g-$C_3N_4$/CQDs nanocomposites under visible-light illumination were delineated in (Figure 4b). The up-conversion photoluminescent (UCPL) properties of CQDs enabled the conversion of longer wavelengths into smaller wavelengths ($\lambda$ = 460 nm), resulting in photo-excitation of charge pairs by g-$C_3N_4$. Subsequently, the matched band configuration imitates electrons migration from CB of g-$C_3N_4$ to CQDs, giving maximum charge carrier separation. Importantly, the photostability of the g-$C_3N_4$/CQDs nanocomposite was also significantly bettered wherein the photocatalytic performance of the photocatalyst endured up to five successive consecutive cycles under light illuminations (Figure 4c). Additionally, a g-$C_3N_4$/CQDs nanocomposite was fabricated through a simple hydrothermal approach and displayed outstanding degradation activity of 71.53% towards 2,4-dichlorophenol, 1.5-fold greater than bare g-$C_3N_4$ [62]. Henceforth, from the above discussion it can be concluded that the best efficacy of the photocatalyst was attributed to CQDs, which acted as electron sinks and hamper the recombination of charge pairs with enhanced separation rate. In addition, UV-visible diffuse reflectance spectrum (UV-vis DRS) analysis detected CQDs as photosensitizers for g-$C_3N_4$, enabling wider-range light harvesting of solar light. This broad-spectrum absorption generates more electrons for the enhancement of photocatalytic performance.

Very recently, a defect-enriched g-$C_3N_4$/N-doped CQDs nanocomposite was synthesized via impregnation approach using urea and citric acid as starting materials by Liu and co-workers [63]. The synthesized binary nanocomposite exhibited excellent light-harvesting capability, narrow band-gap, and feasible electron transport capability, resulting in improved photocatalytic activity towards RhB degradation. Moreover, this binary hybrid displayed the complete RhB elimination rate of 100% along with a $H_2$ generation rate of 3.68 $\mu$mol h$^{-1}$ g$^{-1}$. The g-$C_3N_4$/N-doped CQDs nanocomposite possessed the outstanding electron delocalization feature, which was confirmed using electron paramagnetic resonance (EPR) results (Figure 4d). The spin intensity of the photocatalyst was mainly accelerated with the optimum doping of the N-doped CQDs, revealing the delocalization of single electron and giving enhanced migration and separation efficacy of the electron-hole pairs of the composite. The presence of N-doped CQDs on the surface of defect-rich g-$C_3N_4$ with lattice spacing (100) was observed in a transmission electron microscopy (TEM) image to form hybrid photocatalyst (Figure 4e). Similarly, the S-doped CQDs/g-$C_3N_4$ hybrid was studied for methyl orange degradation with higher sulfur content under visible-light illumination, yielding increased photoactivity [64]. From $N_2$-adsorption–desorption results, we determined the surface area of the photocatalyst and found type-IV adsorption isotherm due to a weak Van der Waals interaction and porous nature (Figure 4f). The excellent surface area was exhibited by the 1:1 S-doped CQDs-10 wt%/g-$C_3N_4$ sample, i.e., 111.72 m$^2$/g, which was 4-fold enhanced to pristine g-$C_3N_4$ (27.58 m$^2$/g). The complete photodegradation efficiency of MO dye was achieved under a UV-visible light range of $\lambda$ > 350 nm within 40 min, while under visible-light irradiation, it reached about 94.3% within 80 min. In other work, a jamun-like designed g-$C_3N_4$/CQDs/$Zn_2V_2O_7$ hybrid composite has been reported for the photodegradation of RhB dye [65]. The Fourier transform infrared (FTIR) spectrum of bare $Zn_2V_2O_7$ and g-$C_3N_4$/CQDs/$Zn_2V_2O_7$ were delineated in (Figure 5a). All featured peaks of bare g-$C_3N_4$ (range of 1235–1650 cm$^{-1}$) and $Zn_2V_2O_7$ (646, 781, and 831 cm$^{-1}$) matched well with the observed peaks in the g-$C_3N_4$/CQDs/$Zn_2V_2O_7$ hybrid, confirming the formation of nanocomposite. However, no peak was observed for the CQDs in the composite, indicating no destruction in the crystallinity and framework of both $Zn_2V_2O_7$ and g-$C_3N_4$ photocatalyst with CQDs deposition during preparation procedure. Additionally, photoluminescence (PL) spectra of as-synthesized g-$C_3N_4$/CQDs/$Zn_2V_2O_7$ hybrid possessed lower recombination rate of charge pairs due to weak emission peak than bare $Zn_2V_2O_7$ and g-$C_3N_4$ photocatalyst as a result of the synergistic effect between the $Zn_2V_2O_7$, g-$C_3N_4$, and CQDs (Figure 5b). Furthermore, electron impedance spectroscopy

(EIS) analysis measured the internal charge transport resistance ($R_{ct}$) on the interfacial contact of bare $Zn_2V_2O_7$ and g-$C_3N_4$/CQDs/$Zn_2V_2O_7$ electrodes and detected smaller $R_{ct}$ for the g-$C_3N_4$/CQDs/$Zn_2V_2O_7$ photocatalyst (Figure 5c). This hybrid photocatalyst exhibited the best conducting nature, which basically enhanced the photodegradation efficiency of RhB (97% in 30 min). In addition, Yang et al., loaded CQDs on the surface of the g-$C_3N_4$ and further combined with $BiVO_4$ forming a g-$C_3N_4$/CQDs/$BiVO_4$ Z-scheme heterostructure through sonication technique under LED light [66]. The g-$C_3N_4$/CQDs/$BiVO_4$-40 wt% Z-scheme exhibited excellent RhB degradation performance (85.1%), compared to the (60.48%) bare g-$C_3N_4$ and (9.6%) $BiVO_4$, which was attributed due to the synergistic effect between each photocatalyst in the Z-scheme. The photostability test showed that negligible activity loss was detected even after the photocatalyst was used for three consecutive cycle experiments.

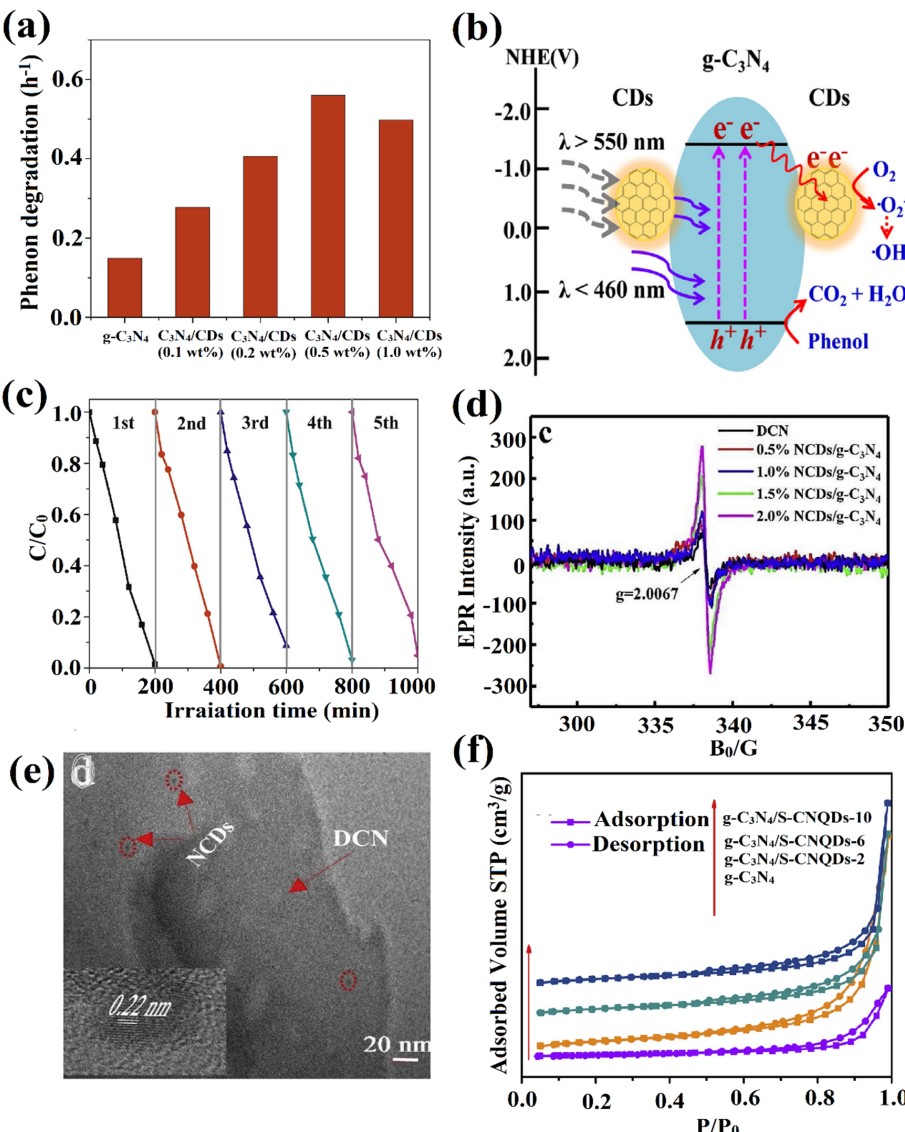

**Figure 4.** (**a**) The rate constant of phenol degradation by different nanocomposite, (**b**) pictorial representation of the possible photocatalytic mechanism of the g-$C_3N_4$/CQDs nanocomposite under visible-light irradiation, (**c**) recyclability test of g-$C_3N_4$/CQDs nanocomposite, (**d**) room temperature electron paramagnetic spectrum, (**e**) 1 wt% N-doped CQDs/defect-rich g-$C_3N_4$, (inset: lattice spacing of 1 wt% N doped CQDs/defect-rich g-$C_3N_4$), and (**f**) $N_2$ adsorption–desorption isotherm at 77 k of S-doped CQDs/g-$C_3N_4$ hybrid composite. Reproduced with permission from Refs. [61,63,64] with license number 5043040500456, 5043050600748, 5043050849974, Copyright Elsevier 2016, 2020.

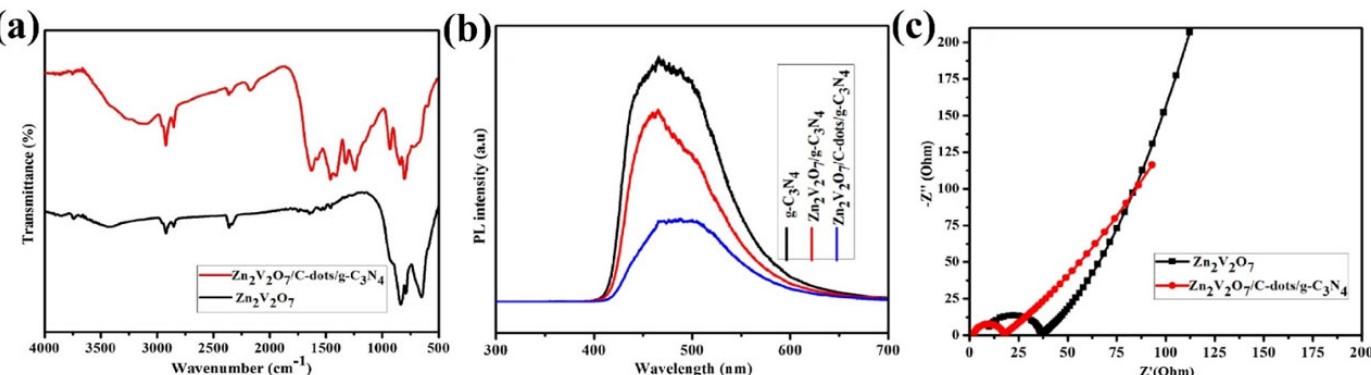

**Figure 5.** (**a**) Fourier transform infrared spectrum, (**b**) time-resolved photoluminescence analysis, and (**c**) electron impedance spectroscopy of $Zn_2V_2O_7$/CQDs/g-$C_3N_4$ hybrid and $Zn_2V_2O_7$. Reproduced with permission from Ref. [65] Copyright Springer 2019.

### 3.2. g-$C_3N_4$/CQDs-Based Nanocomposites for Different Antibiotic Degradation

At the present time, the existence of antibiotics in environmental conditions has received great attention as evolving contaminants, as a result of inappropriate use and effluents discharge from the industrial waste. This causes a serious threat to human and environmental health, primarily attributed to bioaccumulation and hydrophilicity, as well as firm biological stability and activity of antibiotics [67]. Thus, in recent times, an abundance of research data has focused mainly on antibiotic removal from water. For instance, Liu and his peer group reported a g-$C_3N_4$/CQDs composite prepared using thermal polymerization approach towards diclofenac degradation under the visible light of 400 nm [68]. The obtained composite displayed 15-fold enhanced photoactivity in the degradation procedure, in comparison to bare g-$C_3N_4$ as CQDs deposition not only altered the structure and morphology of the g-$C_3N_4$ but also improved the optical absorption range. In order to identify the role of reactive oxidative species in photodegradation of diclofenac, distinct scavengers have been added to the system. With the addition of ammonium oxalate no response was observed in diclofenac degradation, which indicated $h^+$ played no role in degradation process (Figure 6a). On the other hand, when p-benzoquinone and tert-butyl alcohol was added a decreased (first-order) $K_1$ values for diclofenac degradation were detected from 0.14 to $2.7 \times 10^{-3}$ min$^{-1}$ and $9.0 \times 10^{-2}$ min$^{-1}$, respectively. This signified that both $\bullet O_2^-$ and $\bullet OH$ radicals facilitated the degradation procedure and $\bullet O_2^-$ was a dominant species (Figure 6b). In another piece of work, Duan and co-workers reported the novel CQDs decorated g-$C_3N_4$ nanosheet-based nanocomposite, synthesized via facile thermally oxidative etching method for the removal of sulfadiazine [34]. The increase in band gap for the CQDs/g-$C_3N_4$ nanosheet corresponding to the exfoliation temperature was evaluated using fluorescence spectrum, which displayed blue shift emission, attributed from quantum confinement effect of the changed electronic CQDs levels (Figure 6c). Additionally, the transient photocurrent density of the as-synthesized photocatalyst under visible-light irradiation was delineated in (Figure 6d). The CQDs/g-$C_3N_4$ nanosheet (calcined at 500 °C) achieved maximum photocurrent density revealing more effective separation efficiency of the charge pairs than a simple CQDs/g-$C_3N_4$ nanocomposite. In a similar study by Duan et al., the same nanocomposite was fabricated using thermal polymerization approach with in situ growth of g-$C_3N_4$ [69]. The energy dispersive X-ray spectroscopy depicted the N and C atoms in 3 wt% CQDs-g-$C_3N_4$ and an abundance of C was observed, ascribed to in-plane multiple C ring, indicating 3 wt% CQDs-g-$C_3N_4$ were in-plane composite (Figure 6e). Additionally, density functional theory intentions of electronic structure found the in-depth understanding of the relationship between as-prepared photocatalyst and their photoactivity. For a pure g-$C_3N_4$ semiconductor, CB and VB were basically composed of 2p orbitals of C and N, respectively. On the contrary, CQDs/g-$C_3N_4$ possessed smaller band energy with few new energy levels above 0.75 eV, with CB and VB

positioned at CQDs and g-C$_3$N$_4$, respectively (Figure 6f). Thus, this lowering of the band energy resulted into wider-range of light absorptivity, along with efficient separation of charge pairs. As a result, the degradation efficacy of the 3 wt% CQDs/g-C$_3$N$_4$ towards sulfadiazine was reached 92.8% than bare g-C$_3$N$_4$ (53.7% in 80 min).

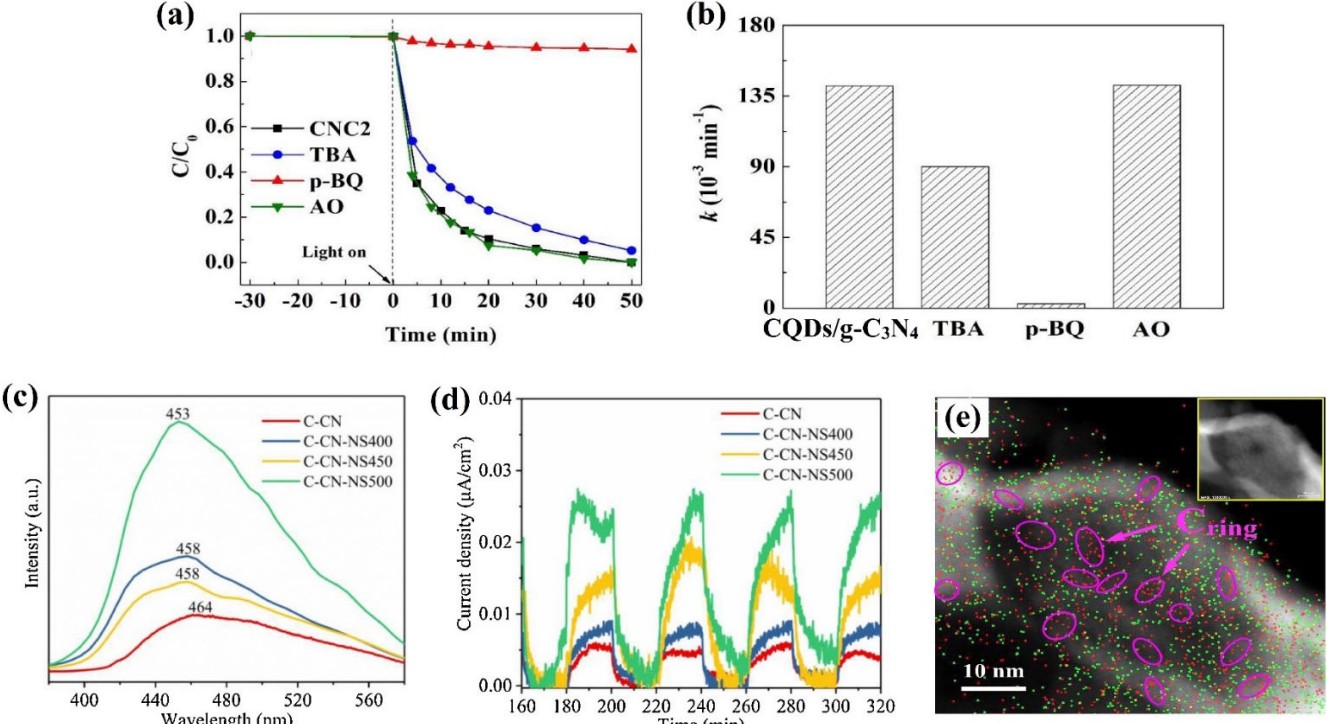

**Figure 6.** (**a**,**b**) Effects of variant scavengers (1 mM ammonium oxalate (AO), 50 μM tert-butyl alcohol, 50 μM, and p-benzoquinone (p-BQ) on the degradation efficiency for diclofenac, (**c**) fluorescence emission spectrum of CQDs/g-C$_3$N$_4$ and different samples (**d**) transient photocurrent densities of CQDs/g-C$_3$N$_4$ and different samples, and (**e**) energy dispersive X-ray spectroscopy of CQDs/g-C$_3$N$_4$ (inset: transmission electron microscopy image of the corresponding region). Reproduced with permission from Refs. [34,68,69] with license number 5043061194671, 5043061353580, 5043070650674, Copyright Elsevier 2019, 2020, and 2021.

Recently, Zhao and co-researchers investigated tubular-shaped g-C$_3$N$_4$-altered CQDs nanocomposite, synthesized by simple polymerization technique for the removal of carbamazepine under light of 400 nm [70]. The photoactivity of the CQDs/g-C$_3$N$_4$ was depicted as being five times greater than the g-C$_3$N$_4$, due to the synergistic effect between the two photocatalyst and strong recyclability for up to four successive cycles. In order to detect the roles of reactive oxidative groups, different scavengers were applied in the experiments, along with electron spin resonance analysis. Here, $^\bullet$O$_2^-$ and h$^+$ were the main species in the degradation process while $^\bullet$OH radicals had no effect. Basically, the K$_1$ values decreased by 98% and 86% by potassium iodide (for h$^+$) and p-benzoquinone (for $^\bullet$O$_2^-$), respectively. On the contrary, only an 8% decrease was observed with the addition of tert-butyl alcohol, employed for quenching $^\bullet$OH species. Furthermore, no signal peak was detected in the electron spin resonance spectra of 5,5-Dimethyl-1-pyrroline N-oxide (DMPO)-$^\bullet$OH species, indicating no $^\bullet$OH formation in the system. Subsequently, DMPO-$^\bullet$O$_2^-$ group showed a strong peak, revealing $^\bullet$O$_2^-$ as a main reactant species. In another study, Shi et al., studied highly crystalline g-C$_3$N$_4$/CQDs nanocomposites, synthesized using calcination techniques, particularly for the degradation of tetracycline [71]. Herein, the possible tetracycline degradation route with product intermediates were recognized using liquid chromatography-mass spectrometry technique. The photoactivity towards tetracycline degradation by g-C$_3$N$_4$/CQDs nanocomposite reached about 86%

within 120 min with the notable photostability of the photocatalyst up to four consecutive cycles with 480 min.

Recently, Liang and his peer group studied the integration of CQDs into acidified g-C$_3$N$_4$ and coupling with CdIn$_2$S$_4$ forming g-C$_3$N$_4$/CQDs/CdIn$_2$S$_4$ photocatalyst and prepared using simple hydrothermal technique [72]. The best photoactivity of hybrid composite towards ibuprofen degradation was found to be 91.0% under 420 nm of visible-light. For optical properties, the strong intense absorption peaks for g-C$_3$N$_4$ and CdIn$_2$S$_4$ at 600 and 680 nm was estimated using UV-visible diffuse reflectance spectroscopy. The absorption range of CdIn$_2$S$_4$ was simultaneously increased by the introduction of CQDs, whereas light absorption spectrum was found to be weaker for ternary g-C$_3$N$_4$/CQDs/CdIn$_2$S$_4$ due to the shielding effect of g-C$_3$N$_4$, along with a band gap of 2.26 eV. Although photostability of the hybrid influences photoactivity, other parameters, such as change migration resistance and charge pairs separation efficiency, also played an important role in the degradation process. Thus, photoluminescence and transient photocurrent density analysis suggested the rate of recombination of charge pairs. The photocurrent responses were found to be greater for g-C$_3$N$_4$/CQDs/CdIn$_2$S$_4$ photocatalysts than bare CdIn$_2$S$_4$, owing to the fast photoexcited transport of charges from CQDs and, finally, the photocatalytic performance was increased. From photoluminescence spectra showed intense peak for CdIn$_2$S$_4$ at 562 nm was attributed to strong recombination between charge pairs, while, with addition of CQDs, the intensity peak was lowered and negligible peak was found for the ternary g-C$_3$N$_4$/CQDs/CdIn$_2$S$_4$ composite. These characteristic techniques results indicated that g-C$_3$N$_4$/CQDs was responsible for the improved charge separation efficiency. Furthermore, Kumar et al., investigated a novel CQDs and reduced graphene oxide (rGO)-altered g-C$_3$N$_4$-CQDs/rGO-g-C$_3$N$_4$ composites for chloramphenicol degradation under solar light irradiation [73]. The optimized g-C$_3$N$_4$-CQDs/rGO-g-C$_3$N$_4$ composite achieved outstanding photocatalytic activity in chloramphenicol degradation, which was about 92.4% within 120 min and 99.1% within 90 min under sunlight and visible-light illumination, respectively. This excellent photoactivity of all the photocatalysts was owed to their sufficient charge pairs separation efficacy through the migration of charge carriers among the band potentials, CQDs and rGO. Here, CQDs had shifted the type-II heterostructure into direct Z-scheme routes and improved the photodegradation efficacy by 10-fold, i.e., 0.0810 min$^{-1}$, in comparison to pristine g-C$_3$N$_4$ 0.00802 min$^{-1}$. Additionally, the amount of $^\bullet$O$_2^-$ and $^\bullet$OH radical generation was increased after Z-scheme formation, rather than individual-doped photocatalysts, which promoted the photocatalytic activity. The overview of g-C3N4/CQDs based photocatalyst for pollutant degradation has been given in Table 1.

**Table 1.** Carbon quantum dots/g-C$_3$N$_4$-based photocatalysts for pollutant degradation under various conditions.

| Type of Composite | Target Pollutant | Source of Illumination Light | Degradation Efficiency | Ref. |
|---|---|---|---|---|
| Carbon-dots/g-C$_3$N$_4$ | Rhodamine B (RhB) dye | 3 W LED lamp (365 ± 5 nm) | RhB degradation: 0.013 mol L$^{-1}$ min$^{-1}$ within 60 min | [60] |
| g-C$_3$N$_4$/Carbon dots (0.5 wt%) | Phenol | 300 W Xe lamp (λ < 400 nm) | Phenol degradation: 100% within 200 min | [61] |
| Carbon-dots/g-C$_3$N$_4$ | 2,4-dichlorophenol (2,4-DCP) | Solar light | 2,4-DCP degradation: 71.53% within 2 h | [62] |
| N-doped Carbon-dots/defect-rich g-C$_3$N$_4$ | RhB dye | 300 W Xe lamp (λ > 420 nm) | RhB degradation: 100% within 4 h | [63] |
| g-C$_3$N$_4$/S-doped carbon nitride quantum dots (S-CNQDs) | Methyl Orange (MO) | 300 W Xe lamp (λ > 350 nm) | MO degradation: 100% within 2 h | [64] |
| Zn$_2$V$_2$O$_7$/Carbon dots/g-C$_3$N$_4$ | RhB dye | 300 W Xe lamp (λ > 420 nm) | RhB degradation: 97% within 30 min | [65] |
| BiVO$_4$/Carbon dots/g-C$_3$N$_4$ | Minocycline hydrochloride (Mn-HCL) and RhB dye | 350 W LED lamp (460 nm) | Mn-HCL degradation: 65.3 within 140 min RhB degradation: 85% within 30 min | [66] |

**Table 1.** *Cont.*

| Type of Composite | Target Pollutant | Source of Illumination Light | Degradation Efficiency | Ref. |
|---|---|---|---|---|
| Carbon quantum dots/g-$C_3N_4$ | Diclofenac (DCF) | 300 W Xe lamp ($\lambda$ > 420 nm) | Diclofenac degradation: 100% within 60 min | [68] |
| g-$C_3N_4$/Carbon-dots nanosheet | Sulfadiazine (SDZ) | 500 W Xe lamp ($\lambda$ < 420 nm) | SDZ degradation: 97.82% within 60 min | [34] |
| Carbon-dots/g-$C_3N_4$ | SDZ | Visible-light | SDZ degradation: 92.8% within 80 min | [69] |
| Carbon quantum dots/g-$C_3N_4$ | Carbamazepine (CBZ) | UV-light with cutoff filter ($\lambda$ < 400 nm) | CBZ degraded with first-order rate constant of $5.73 \times 10^{-2}$ $min^{-1}$ | [70] |
| High-crystalline g-$C_3N_4$/Carbon-dots | Tetracycline (TC) | 300 W Xe lamp ($\lambda$ > 420 nm) | TC degradation: 86% within 120 min | [71] |
| Acidified g-$C_3N_4$/Carbon quantum dots-3 wt%/$CdIn_2S_4$ | Ibuprofen (IBU) | 300 W Xe lamp ($\lambda$ > 420 nm) | IBU degradation: 91.0% within 60 min | [72] |
| S@g-$C_3N_4$/B@B-g-$C_3N_4$ | Chloramphenicol (CMP) | 300 W Xe lamp with cutoff filters (380–780 nm) | CMP degradation: 99.1% within 1 h | [73] |

### 3.3. g-$C_3N_4$/CQDs-Based Nanocomposites for $CO_2$ Reduction

Photocatalytic $CO_2$ conversion into solar fuels using sunlight offers a tempting pathway to lessen environmental issues (such as climate change) and global energy [74]. Amid the greenhouse gases, $CO_2$ has been represented as the most prominent member of the group; hence, offering treatment towards resolving this issue seems to be of supreme importance [75,76]. As a result, many researchers have focused in the most recent years on the reduction in inorganics, specifically the $CO_2$ molecule because of the aforementioned environmental problems [77]. Ultimately, hydrocarbon production from $CO_2$ conversion has been considered the potential solution to the problems of environmental catastrophes. Hence, many researchers have found metal-free photocatalysts, such as g-$C_3N_4$, photosensitized by CQDs because of their sensitizing feature and excellent conducting nature. Very recently, Jiang et al., reported g-$C_3N_4$ CQDs deposited mesoporous $CeO_2$ nanocomposite synthesized through simple hydrothermal technique [78]. The diffuse reflectance spectrum revealed that mesoporous $CeO_2$ exhibited light absorption up to range of 410 nm while with the deposition of g-$C_3N_4$-CQDs to the $CeO_2$ surface converted the absorptivity region to 422 nm. The surface area of the (0.75 wt%) g-$C_3N_4$-CQDs/$CeO_2$ (147.15 $m^2$/g) was found to be lower than $CeO_2$ (150.79 $m^2$/g) due to the presence of quantum dots. The photoactivity was determined for $CO_2$ conversion into $CH_4$ and CO, yielding 15.81 $\mu$mol/g and 22.48 $\mu$mol/g, respectively, under UV-light illumination for 10 h. In other work, Zhao and co-workers synthesized g-$C_3N_4$/CQDs/Au composite through a facile one-step photochemical approach and employed further for the photoconversion of $CO_2$ [79]. This composite possessed a significant CO faradic efficacy of 79.8% at potential of 0.5 V. This enhanced photoactivity of the photocatalyst was owed to the synergistic relation between the Au, CQDs, and g-$C_3N_4$ nanomaterials. Additionally, the combination of g-$C_3N_4$ with CQDs could simplify the $CO_2$ conversion into CO because of excellent conducting nature of CQDs, $CO_2$ adsorption, and superior capability of $H^+$. The ultra-high surface area of the hybrid photocatalyst, i.e., 117 $m^2$ $g^{-1}$ was the actual reason for the extreme activeness of the photocatalyst towards CO production.

## 4. Conclusions

Keeping in mind the sustainable growth of human life, contaminant-free approaches are of crucial necessity. Although substantial efforts have been employed to construct variant types of photocatalysts, minimal research data have been reported to solve these problems comprehensibly, among which g-$C_3N_4$/CQDs-based photosystems have been studied and explored extensively. Due to modified electronic structure, CQDs-decorated g-$C_3N_4$ photosystems have become fast growing field of research. This article initially provides the designing strategies and physicochemical properties of g-$C_3N_4$/CQDs-based

nanocomposites. The review mainly summarized the g-$C_3N_4$/CQDs-based nanocomposites in the field of environmental remediation, including the degradation of dyes and different antibiotics, and $CO_2$ conversion into valuable chemicals. The g-$C_3N_4$/CQDs-based photocatalysts were mainly explored for dye and antibiotic degradation, as per existing literature reports, indicating their effectiveness in the wastewater remediations. Still, the formation of harmful secondary by-products during photodegradation processes needs major consideration to avoid such pollutant products. The development of effective methods to load CQDs on g-$C_3N_4$-based nanocomposites and to enhance the recyclability of g-$C_3N_4$/CQDs heterostructure are also crucial parameters. This g-$C_3N_4$-based heterostructuralization could enhance the light-harvesting capability, charge kinetic reactions, and their separation efficiency, ultra-high surface area, and photochemical stability, giving promoted photoactivity of the photocatalyst. However, despite the aforementioned advantages and achievements in the wastewater treatment, there still exists some ongoing challenges and problems that hampers the practical applications of the g-$C_3N_4$/CQDs-based nanocomposite:

i.     Even though there is prodigious development in the modified g-$C_3N_4$-based composite, the visible-light utilization is still limited to near blue-violet spectrum range, and subsequently, the efficacy of light-harvesting and solar-light employment is not high. However, the g-$C_3N_4$/CQDs nanocomposites are only limited to laboratory scale due to photocatalyst dissolution during photoreaction and corrosion of the photocatalyst. Hence, future research should focus more on developing modified g-$C_3N_4$ by constructing more stable structural framework and stronger bonds between the components. For instance, in thermal polymerization approach followed through the conversion of monomers into polymeric precursors under a high-temperature environment, forming strong hydrogen bonds amid precursors, resulting into enhanced connection between g-$C_3N_4$ and CQDs.

ii.     The green fabrication approaches of the g-$C_3N_4$/CQDs hybrid composite with excellent photochemical stability, intrinsic electronic structure, efficient photoactivity, and reusability with potential scalability are still in their infancy stage. Thus, further research must be devoted to employing more efficient synthesis processes without affecting any of the above-mentioned parameters.

iii.     The charge transport pathway in distinct g-$C_3N_4$/CQDs-based nanocomposites, as well as photocatalytic mechanism, need to be explored. Basically, the photo-assisted performance of g-$C_3N_4$ was optimized with QDs, which effectively captured the photoexcited electrons in the g-$C_3N_4$. In addition, a deep understanding of the synergistic effect between g-$C_3N_4$ and CQDs interfaces is still lacking. In light of this, in situ test techniques and theoretical calculations and their experimental results are predictable to help researchers to understand the mystery of the charge migrating route.

iv.     Although there are an abundance of reports on g-$C_3N_4$/CQDs-based nanocomposites for different types of pollutant degradation, they are still in their infancy, specifically concerning $CO_2$ conversion with some indefinite reduction product challenges, imprecise number of generated products, and inappropriate charge migration route. The future research should focus on $CO_2$ mitigation using this composite and also on innovative experiments and computation techniques to have enough knowledge of the reaction mechanism and migration route from thermodynamic and kinetic view point.

**Author Contributions:** S.P.: Investigating, methodology, writing—review & editing. S.: Data curation, writing—review & editing. A.S.: Investigating, writing—review & editing. N.C. (Naresh Chandel): Writing—review & editing, T.A.: Writing—review & editing. P.R.: Conceptualization, writing—original draft, formal analysis. P.S.: Formal analysis, supervision. N.C. (Nhamo Chaukura): Writing—review & editing. R.S. Conceptualization, writing—original draft, formal analysis. All authors have read and agreed to the published version of the manuscript.

**Funding:** This research received no external funding.

**Conflicts of Interest:** The authors do not have any conflict of interest.

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
