# Peer review of "A Review on Carbon Quantum Dots Modified g-C3N4-Based Photocatalysts and Potential Application in Wastewater Treatment"

_applsci, doi:10.3390/app122111286_

Round 1

Reviewer 1 Report

The comments and suggestions for improving the work can be found on the suggestions document attached.

Author Response

The manuscript work entitled “A review on carbon quantum dots modified g-C3N4-based photocatalysts and potential application in wastewater treatment” is interesting and up to date, however, I felt the necessity of some adjustments to improve the quality of the paper. If the authors could address properly the comments stated below, I consider appropriate the work for publication.

Q1. In the sentence “…researchers on TiO2 photocatalysis [10]. Thereafter….” I suggest to expand better the outreach of citations more than only a single one. Photocatalysis authors i.e. Malato et. Al. in Spain and its group are excellent researchers in the wastewater treatment and in works also harvesting sunlight to empower the photocatalytic process. The same is to be referred on semiconductors different than TiO2, this certainly can be expanded.

Response: Dear Review, we agree with your comment and therefore added the suggested references with discussion in order to certainly expand the discussion. Also, a citation of related work to TiO2 is added from the journal of Applied Sciences.

  • Please refer ref. 16, 17, and, 37.

Q2. Regarding figure 1, I suggest to expand in the introduction explanations with more details and with proper citations each of the criteria for a “good photocatalyst”.

Response: Dear Reviewer, we have expanded the mentioned explanation with more details and proper citations each of the criteria for a “good photocatalyst”.

  • Please refer page no.- 3, section Introduction

Q3. “…Therefore, the g-C3N4/CQDs based nanocomposites have some diverse merits over the bare g-C3N4 such as broader visible-light absorption, ultra-high specific surface area, and enhanced electron-hole pair migration and separation efficiency….” Lacks a citation on the sentence.

Response: Dear Reviewer as per your suggestion we have incorporated the relevant citation on the mentioned sentence in the revised manuscript.

  • Please refer page no.- 6, ref. [53].

Q4. Figures 2 and 3, if they are not designed by the authors, it should be asked the rights permission and stated on the labels.

Response: Dear Reviewer, as per your valuable comment we have added the permission of the rights and stated on the captions of figures 1 and 2 (designed one), and for figure 3 the rights permissions are already mentioned.

  • Please refer ref. 31 for fig. 1 and 23 for fig. 3 (in present revised manuscript).

Q5. In order to clarify better the results and developments regarding the CQD/C3N4 systems, please create a table maybe in the end of the document with the main results stated along the text. In this table the columns could be organized by: reference (citation of the work), type of catalyst (composite obtained), pollutant studied, illumination system on photocatalysis (wavelength, artificial, natural, power, etc.), and the yields of degradation results with the optimal time. This can help the reader to have a clear picture of the actual point the nanocomposite system is.

Response: Dear Review, we have added a table in the end of the document with the main results stated along the text and highlighted in the revised manuscript.

  • Please refer Table, page number- 17

Table: Carbon quantum dots/g-C3N4 based photocatalysts for pollutant degradation under various conditions.

Type of composite

Target pollutant

Source of illumination light

Degradation efficiency

Ref.

Carbon-dots/g-C3N4

Rhodamine B (RhB) dye

3W LED lamp (365 ± 5 nm)

RhB degradation: 0.013 mol L-1 min-1 within 60 minutes

58

g-C3N4/Carbon dots (0.5 wt%)

Phenol

300 W Xe lamp (λ< 400 nm)

Phenol degradation: 100% within 200 minutes

59

Carbon-dots/g-C3N4

2,4-dichlorophenol (2,4-DCP)

Solar light

2,4-DCP degradation: 71.53% within 2h

60

N-doped Carbon-dots/defect-rich g-C3N4

RhB dye

300 W Xe lamp (λ> 420 nm)

RhB degradation: 100% within 4 h

61

g-C3N4/S-doped carbon nitride quantum dots (S-CNQDs)

Methyl Orange (MO)

300 W Xe lamp (λ> 350 nm)

MO degradation: 100% within 2h

62

Zn2V2O7/Carbon dots/ g-C3N4

RhB dye

300 W Xe lamp (λ> 420 nm)

RhB degradation: 97% within 30 minutes

63

BiVO4/Carbon dots/g-C3N4

Minocycline hydrochloride (Mn-HCL) and RhB dye

350 W LED lamp (460 nm)

Mn-HCL degradation: 65.3 within 140 min

RhB degradation: 85% within 30 minutes

64

Carbon quantum dots/g-C3N4

Diclofenac (DCF)

300 W Xe lamp (λ> 420 nm)

Diclofenac degradation: 100% within 60 minutes

66

g-C3N4/ Carbon-dots nanosheet

Sulfadiazine (SDZ)

500 W Xe lamp (λ< 420 nm)

SDZ degradation: 97.82% within 60 minutes

67

Carbon-dots/g-C3N4

SDZ

Visible-light

SDZ degradation: 92.8% within 80 minutes

68

Carbon quantum dots/g-C3N4

Carbamazepine (CBZ)

UV-light with cutoff filter (λ< 400 nm)

CBZ degraded with first-order rate constant of 5.73 × 10-2 min-1

69

High-crystalline g-C3N4/ Carbon-dots

Tetracycline (TC)

300 W Xe lamp (λ> 420 nm)

TC degradation: 86% within 120 minutes

70

Acidified g-C3N4/Carbon quantum dots-3wt %/CdIn2S4

Ibuprofen (IBU)

300 W Xe lamp (λ> 420 nm)

IBU degradation: 91.0 % within 60 minutes

71

S@g-C3N4/B@B-g-C3N4

Chloramphenicol (CMP)

300 W Xe lamp with cutoff filters (380-780 nm)

CMP degradation: 99.1% within 1h

72

Q6. As a final comment, a review article to be well supported has normally between 100-150 citations, otherwise the reader does not have the proper idea of the deep study performed and understanding the authors are trying to express. I would recommend revising all non-cited sentences/paragraphs from the text and refer the original source providing the citation from where each statement comes from.

Response: Dear Reviewer, thank you for your valuable comment, we have added the citations as per your recommendation in the sentences/paragraphs and tried our best to extend the citations to 80 with relevant citations in the revised manuscript.

Reviewer 2 Report

Dear Editor

This study investigated a review based on carbon quantum dots modified g-C3N4-based photocatalysts for wastewater application.

1.     Manuscript needs references from applied sciences journal

2.     Equations need references

3.     Introduction needs a curve “growth of number publications as a function of year for photocatalysts for wastewater”

4.     In Fig. 2. Schematic representation of the role of CQDs in the g-C3N4/CQDs based nanocomposite with improved photocatalytic performance…. (a) to (d) were not defined.

5.     Please compare references of “section 3.1. g-C3N4/CQDs based nanocomposites for pollutant degradation” in a table (  photoactivity, wavelengths, …)

6.     Section “3.3. g-C3N4/CQDs based nanocomposites for CO2 reduction” need more references about CO2 reduction

7.     In the conclusion section, please explain the results of compared references (sections, 3.1, 3.2, and 3.3)

8.     Even though the prodigious development in the modified g-C3N4 based composite, the visible-light utilization is still limited to near blue-violet spectrum range, and subsequently, the efficacy of light-harvesting and solar-light employment is not high. However, the g-C3N4/CQDs nanocomposites are only limited to laboratory scale due to photocatalyst dissolution during photoreaction and corrosion of the photocatalyst. Hence, future research should focus more on developing modified g-C3N4 by “constructing more stable structural framework and stronger bonds between the components”….. Please more explain and propose an approach for “stronger bonds between the components”

9.     The charge transport pathway in distinct g-C3N4/CQDs based nanocomposite as well as photocatalytic mechanism need to be explored….please more explain.

10.  References are not matching the format

Author Response

This study investigated a review based on carbon quantum dots modified g-C3N4-based photocatalysts for wastewater application.

Q1. Manuscript needs references from applied sciences journal

Response: Dear Reviewer, as per your requirement the references from the applied sciences journal are added in the revised manuscript.

  • Please refer ref. 4-9, 37, 38, 75

Q2. Equations need references

Response: Dear Reviewer, we have added the reference for the equation in the revised manuscript.

  • Please refer ref. 39

Q3.     Introduction needs a curve “growth of number publications as a function of year for photocatalysts for wastewater”.

Response: Dear Reviewer, as per your requirement we have incorporated bar graph for “growth of number publications as a function of year for CQDs/g-C3N4 based photocatalysts for wastewater” in the revised manuscript.

Fig. 2. Overall publication detail on CQDs/g-C3N4 based photocatalysts for wastewater treatment from 2013 to 2022.

Q4. In Fig. 2. Schematic representation of the role of CQDs in the g-C3N4/CQDs based nanocomposite with improved photocatalytic performance…. (a) to (d) were not defined.

Response: Dear Reviewer, thank you for pointing out this, we have incorporated the role of CQDs in the g-C3N4/CQDs based nanocomposite in the caption of fig. 3 and highlighted in the revised manuscript.

  • Please refer Fig. 3. caption

Fig. 3. Schematic representation of the role of CQDs in the g-C3N4/CQDs based nanocomposite, (a) CQDs hampers electron-hole pair recombination, (b) CQDs enables photoexcitation of electrons even with 550 nm incident light, CQDs serve as electron mediator in (c) type-II, and (d) Z-schemes. Adapted from ref. 1.

Q5. Please compare references of “section 3.1. g-C3N4/CQDs based nanocomposites for pollutant degradation” in a table (photoactivity, wavelengths, …).

Response: Dear Reviewer as per your valuable suggestion we have introduced the table at the end of the section “3.1. g-C3N4/CQDs based nanocomposites for pollutant degradation” in the revised manuscript.

Table: Carbon quantum dots/g-C3N4 based photocatalysts for pollutant degradation under various conditions.

Type of composite

Target pollutant

Source of illumination light

Degradation efficiency

Ref.

Carbon-dots/g-C3N4

Rhodamine B (RhB) dye

3W LED lamp (365 ± 5 nm)

RhB degradation: 0.013 mol L-1 min-1 within 60 minutes

58

g-C3N4/Carbon dots (0.5 wt%)

Phenol

300 W Xe lamp (λ< 400 nm)

Phenol degradation: 100% within 200 minutes

59

Carbon-dots/g-C3N4

2,4-dichlorophenol (2,4-DCP)

Solar light

2,4-DCP degradation: 71.53% within 2h

60

N-doped Carbon-dots/defect-rich g-C3N4

RhB dye

300 W Xe lamp (λ> 420 nm)

RhB degradation: 100% within 4 h

61

g-C3N4/S-doped carbon nitride quantum dots (S-CNQDs)

Methyl Orange (MO)

300 W Xe lamp (λ> 350 nm)

MO degradation: 100% within 2h

62

Zn2V2O7/Carbon dots/ g-C3N4

RhB dye

300 W Xe lamp (λ> 420 nm)

RhB degradation: 97% within 30 minutes

63

BiVO4/Carbon dots/g-C3N4

Minocycline hydrochloride (Mn-HCL) and RhB dye

350 W LED lamp (460 nm)

Mn-HCL degradation: 65.3 within 140 min

RhB degradation: 85% within 30 minutes

64

Carbon quantum dots/g-C3N4

Diclofenac (DCF)

300 W Xe lamp (λ> 420 nm)

Diclofenac degradation: 100% within 60 minutes

66

g-C3N4/ Carbon-dots nanosheet

Sulfadiazine (SDZ)

500 W Xe lamp (λ< 420 nm)

SDZ degradation: 97.82% within 60 minutes

67

Carbon-dots/g-C3N4

SDZ

Visible-light

SDZ degradation: 92.8% within 80 minutes

68

Carbon quantum dots/g-C3N4

Carbamazepine (CBZ)

UV-light with cutoff filter (λ< 400 nm)

CBZ degraded with first-order rate constant of 5.73 × 10-2 min-1

69

High-crystalline g-C3N4/ Carbon-dots

Tetracycline (TC)

300 W Xe lamp (λ> 420 nm)

TC degradation: 86% within 120 minutes

70

Acidified g-C3N4/Carbon quantum dots-3wt %/CdIn2S4

Ibuprofen (IBU)

300 W Xe lamp (λ> 420 nm)

IBU degradation: 91.0 % within 60 minutes

71

S@g-C3N4/B@B-g-C3N4

Chloramphenicol (CMP)

300 W Xe lamp with cutoff filters (380-780 nm)

CMP degradation: 99.1% within 1h

72

Q6. Section “3.3. g-C3N4/CQDs based nanocomposites for CO2 reduction” need more references about CO2 reduction

Response: Dear Reviewer as per your requirement we have incorporated the relevant references in the section “3.3. g-C3N4/CQDs based nanocomposites for CO2 reduction”.

  • Please refer ref. 75 to 78

Q7. In the conclusion section, please explain the results of compared references (sections, 3.1, 3.2, and 3.3)

Response: Dear Reviewer, as per your requirement, we have incorporated the more comparative explanation of the results discussed in the reference (sections, 3.1, 3.2, and 3.3).

This article initially provides the designing strategies and physicochemical properties of g-C3N4/CQDs based nanocomposites. The review mainly summarised the g-C3N4/CQDs based nanocomposites in the field of environmental remediation including degradation of dyes and different antibiotics, and CO2 conversion into valuable chemicals. The g-C3N4/CQDs based photocatalysts were mainly explored for dye and antibiotic degradation as per existing literature reports, indicating its effectiveness in the wastewater remediations. Still, the formation of harmful secondary by-products during photodegradation process needs major consideration to avoid such pollutant products. Development of effective methods to load CQDs on g-C3N4-based nanocomposites and to enhance the recyclability of g-C3N4/CQDs heterostructure are also crucial parameter. This g-C3N4 based hetero-structuralization could enhance the light-harvesting capability, charge kinetic reactions and their separation efficiency, ultra-high surface area, and photochemical stability, giving promoted photoactivity of the photocatalyst.

Q8. Even though the prodigious development in the modified g-C3N4 based composite, the visible-light utilization is still limited to near blue-violet spectrum range, and subsequently, the efficacy of light-harvesting and solar-light employment is not high. However, the g-C3N4/CQDs nanocomposites are only limited to laboratory scale due to photocatalyst dissolution during photoreaction and corrosion of the photocatalyst. Hence, future research should focus more on developing modified g-C3N4 by “constructing more stable structural framework and stronger bonds between the components”….. Please more explain and propose an approach for “stronger bonds between the components”.

Response: Dear Reviewer, thank you for your valuable comment, we have incorporated the more relevant explanations to the above-mentioned point and proposed an approach for stronger bonds between CQDs and g-C3N4.

The relevant discussion in the revised text is as follows:

  • Even though the prodigious development in the modified g-C3N4 based composite, the visible-light utilization is still limited to near blue-violet spectrum range, and subsequently, the efficacy of light-harvesting and solar-light employment is not high. However, the g-C3N4/CQDs nanocomposites are only limited to laboratory scale due to photocatalyst dissolution during photoreaction and corrosion of the photocatalyst. Hence, future research should focus more on developing modified g-C3N4 by constructing more stable structural framework and stronger bonds between the components. For instance, in thermal polymerization approach followed through the conversion of monomers into polymeric precursors under high temperature environment forming strong hydrogen bonds amid precursors, resulting into enhanced connection between g-C3N4 and CQDs.

Q9. The charge transport pathway in distinct g-C3N4/CQDs based nanocomposite, as well as photocatalytic mechanism, need to be explored….please more explain.

Response: Dear Reviewer, as per your valuable comment we have added more discussion for the charge transport route followed in g-C3N4/CQDs based nanocomposite.

The charge transport pathway in distinct g-C3N4/CQDs based nanocomposite, as well as photocatalytic mechanism, need to be explored. Basically, the photoassisted performance of g-C3N4 was optimized with QDs which effectively captured the photo-excited electrons in the g-C3N4.

Q10.  References are not matching the format

Response: Dear Reviewer, thank you for your valuable comments, we have checked the format of Applied Sciences and corrected it according to the present journal in the revised manuscript.

Round 2

Reviewer 1 Report

I could observe the suggestions of modifications were accepted and therefore I recommend the publication of the work.

Reviewer 2 Report

Dear Editor 

The manuscript well has been revised.